



# Massive carbon addition to an organic-rich Andosol did not increase the topsoil but the subsoil carbon stock.

Antonia Zieger[1], Klaus Kaiser[2], Pedro Ríos Guayasamín[3], and Martin Kaupenjohann[1]

[1]Department of Soil Science, Institute of Ecology, Technische Universität Berlin, Ernst-Reuter-Platz 1, 10587 Berlin, Germany
[2]Soil Science and Soil Protection, Institute of Agricultural and Nutritional Science, Martin Luther University, Von-Seckendorff-Platz 3, 06120 Halle (Saale), Germany
[3]Laboratorio de Ecología Tropical Natural y Aplicada, Universidad Estatal Amazónica, Campus Principal Km 2.1/2 via a Napo (Paso Lateral) Puyo, Pastaza, Ecuador

*Correspondence to:* Antonia Zieger (antonia.zieger@tu-berlin.de)

**Abstract.** Andosols are among the most carbon rich soils, with an average of 254 Mg ha$^{-1}$ organic carbon (OC) in the upper 100 cm. A current theory proposes an upper limit for OC stocks independent of increasing carbon input. This is assigned to finite binding capacities for organic matter (OM) of the soil mineral phase. We tested the possible limits in OC stocks for Andosols with already large OC concentrations and stocks (210 g kg$^{-1}$ in the first horizon; 320 Mg ha$^{-1}$ in the upper 100 cm).

The soils received large inputs of 1800 Mg OC ha$^{-1}$ as sawdust within a time period of 20 years. Adjacent soils without sawdust application served as controls. We determined total OC stocks as well as the storage forms of OM of five horizons down to 100 cm depth. Storage forms considered were pyrogenic carbon, OM of < 1.6 g cm$^{-3}$ density and with no to little interaction with the mineral phase, strongly mineral-bonded OM forming particles of densities between 1.6 and 2.0 g cm$^{-3}$ or > 2.0 g cm$^{-3}$. The two fractions > 1.6 g cm$^{-3}$ were also analyzed for Al-organic matter complexes (Al-OM complexes) and

imogolite-type phases using ammonium oxalate-oxalic acid extraction and X-ray diffraction (XRD).

Pyrogenic organic carbon represented only up to 5 wt% of OC, and thus, contributed little to soil OM. In the two topsoil horizons, the fraction between >1.6 and 2.0 g cm$^{-3}$ had 65-86 wt% of bulk soil OC and were dominated by Al-OM complexes. In deeper horizons, the fraction > 2.0 g cm$^{-3}$ contained 80-97 wt% of bulk soil's total OC and was characterized by a mixture of Al-OM complexes and imogolite-type phases, with proportions of imogolite-type phases increasing with depth. In response

to the sawdust application, only the OC stock in 25-50 cm depth increased significantly (P = 0.05). The increase was entirely due to increased OC in the two fractions > 1.6 g cm$^{-3}$. However, there was no significant increase in the total OC stocks within the upper 100 cm.

We assume, the topsoil is saturated in terms of OC concentrations, and thus, added OC partly migrates downwards, where it becomes retained by OC-undersaturated minerals. This indicates the possibility to sustainably increase already large OC stocks

further, given that the subsoil still has binding capacity and OC transport into deeper horizons is facilitated. The little additional OC accumulation despite the extremely large OC input over 20 years, however, shows that long time periods of high input are needed to promote the downward movement and deep soil storage of OC.

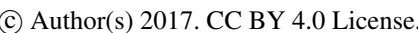



# 1 Introduction

Soil holds more organic carbon (OC) than there is carbon in the global vegetation and atmosphere combined. Moreover soil organic matter (OM) improves plant growth and protects water quality by retaining nutrients as well as pollutants in the soil

(Lal, 2004). Thus, understanding the soil OC dynamics is crucial for developing strategies to mitigate the increase of atmospheric $CO_2$ concentrations and increase soil fertility (Stewart et al., 2007). The large soil carbon reservoir, however is not steady. It results from a dynamic equilibrium between organic and inorganic material entering and leaving the soil (Schrumpf et al., 2011).

There are contradictory views on soil carbon storage capacities. According to Lal (2004), the OC stock to 1 m depth ranges from 30 in arid climates to $800 \, \mathrm{Mg \, ha^{-1}}$ in organic soils in cold regions; the predominant range is 50 to $150 \, \mathrm{Mg \, ha^{-1}}$. Paustian et al. (1997) consider the carbon input rate as the main factor and state that the OC stocks increase linearly with increasing organic input without having an upper limit. Most current OC models, which use this linear relationship, perform reasonably well across a diversity of soils and land use changes (Campbell and Paustian, 2015). On the contrary, Campbell et al. (1991)

published data, where soils rich in OC show little or no increase in soil OC despite a two to three fold increase in carbon input. This motivated Six et al. (2002) and Stewart et al. (2007) to propose that OC accumulation potentials of soils are limited independent of increasing carbon input. The authors attribute this to the limited binding capacities of minerals. This concept is reflected by the model of Schmidt et al. (2011), in which the OC input is stepwise mineralized, surpassing the form of large biopolymeres, small biopolymers with less than 600 Da and monomers. At each step the possibility of interaction with mineral

phases increases, leading to different OC storage forms with differing turnover times and degree of interaction with the mineral phase. Overall the predominant proportion of OM in soils is associated with the mineral phase (e.g. Schrumpf et al., 2013). Minerals have finite reactive surface areas and consequently finite OM binding capacities. The size of the surface area depends on the type of mineral, and so, the differences in OC stocks among soils are due to different types and amounts of the contained minerals. Thus, the OC input rate is only crucial as long as the mineral OC storage capacities are not exhausted. However, the

concept of limited storage capacity has hardly been experimentally tested so far.

Allophane and imogolite-type phases are, besides Al and Fe oxides, the most effective minerals to bind OM (Kögel-Knabner et al., 2008; Huang et al., 2011a). They dominate the mineral assemblage of Andosols, making them the most carbon-rich mineral soil (Huang et al., 2011a; WRB, 2006; Basile-Doelsch et al., 2007). Andosols are subdivided in silandic and aluandic

subgroups. Silandic Andosols have $80\text{-}120 \, \mathrm{g \, OC \, kg^{-1}}$ soil, whereas aluandic Andosols can contain up to $300 \, \mathrm{g \, OC \, kg^{-1}}$ soil (Huang et al., 2011a). Differences in OC concentrations among both subgroups are explained by differing carbon storage mechanisms. Organic matter in silandic Andosols is mainly bound to allophanes, imogolites and protoimogolites (grouped as imogolite-type phases Levard et al. (2012)). The OM in aluandic Andosols is mainly stored with aluminium-organic complexes





(Al-OM complexes). The Al in these complexes can be either monomeric $Al^{3+}$ ions, but also hydroxilated Al species (Huang et al., 2011a; Colombo et al., 2004; Masion et al., 1994). Independent of the type of Andosol, the main OM stabilizing mechanisms are organic-mineral interactions. Andosols with extremely high OC concentrations likely present OM-saturated mineral phases, at least in the topsoil, and should respond with no change in OC concentrations to increasing carbon input.

In order to test the concept of limited OC storage capacity in soils we took the opportunity of a unique setting in the Ecuadorian rainforest, where a carbon-rich Andosol ($350\,Mg\,OC\,ha^{-1}$ within the first 100 cm) received an extra $1800\,Mg\,OC\,ha^{-1}$ input as sawdust during a period of 20 years. Adjacent soils without sawdust application served as controls.

We tested the following hypotheses: i) The additional OC input did not result in increased OC in the topsoil, but in the subsoil, because the mineral binding capacities for OM in the topsoil are exhausted and mobile OM is transported into the subsoil and retained there; ii) the increase of OC in the subsoil is due to OM binding to the mineral phase; and iii) the total OC stock of the soil increased significantly.

We determined total OC stocks as well as the storage forms of OM and the mineral composition down to 100 cm depth. For determining different OM storage forms we used the sequential density fractionation method to obtain fractions yielding OM with different degrees of mineral interaction. We also determined pyrogenic organic carbon (PyC), because of its significant contribution to the OC stocks in some regions of the Amazon basin (e.g. Glaser et al., 2000)). We used ammonium oxalate–oxalic acid extraction and X-ray diffraction for characterizing the prevalent mineral species in the density fractions containing

20 organic-mineral associations.

## 2 Materials and methods

### 2.1 Soil sample source and handling

The study site is located in Ecuador, within the *Centro de Rescarte de la Flora Amazónica* (CERFA) 3 km south of Puyo (1°30'50" S, 77°58'50" E, 950 AMSL). Puyo, located in the transition zone between the Andes and the western Amazon basin,

lies in the center of an alluvial fan affected by deposition of Pleistocene volcanic debris called the Mera formation (Sauer, 1971). The deposited material belongs to the andesite-plagidacite series or the andesite andesitedacite-rhyolite series. Later on additional thick layers of volcanic ash from eruptions were deposited (Hörmann and Pichler, 1982). Today fresh ash is deposed only infrequently from the volcanoes Tungaraghua and Sangay (Le Pennec et al., 2012). Tungaraghua ash composition ranges from basic andesites to dacites (Hall et al., 1999). The climate is diurnal tropical with mean annual temperatures of 20.8 °C

and annual precipitation of 4403 mm (Schwarz, 2015). The vegetation cover is tropical rainforest and pasture (Tello, 2014).

Before 1980 the sampling area has undergone traditional shifting cultivation and later on pasture dairy farming. Since 1980 7 ha of the pasture were reforested by the individual Nelson Omar Tello Benalcázar. On 3 ha, within this area, he applied





1800 t OC ha$^{-1}$ additional litter in form of sawdust until the year 2000 (sawdust site). About 10 m$^3$ of sawdust where applied manually on 5 days a week for 20 years. The sawdust was collected on a daily basis from a local sawmill. Nowadays the 7 ha are covered by about 37 year old secondary rainforest. (Tello, 2014)

We got in contact with this interesting project in 2013. As the site was originally not designed for experimental purposes it does not reflect a randomized plot design. Nevertheless, we think that it can be scientifically evaluated because the plot area on which we sampled is large (2-3 ha at each site) and essential conditions like exposition, inclination, climate, weather conditions and geology are comparable between the treated and untreated areas. No information about changes in treespecies over time and possible differences between species due to the sawdust input were available. Instead we estimated additional litter input at

the sawdust site due to higher vegetation biomass productivity based on the literature. Clark et al. (2001) reported fine litterfall biomasses of 0.9-6 Mg ha$^{-1}$ year$^{-1}$ for tropical forest all over the world. If the sawdust site would have been covered with a tropical forest for the whole 37 years and the adjacent site (contorl site) not, 222 Mg ha$^{-1}$ litter biomass would have been additionally added since 1980. This litter carbon represents less than 6 % of the total sawdust carbon input and is therefore insignificant. In order to eliminate the belowground biomass as the dominating soil organic carbon source we measured the

gravimetric root intensity. The results show no significant difference between the sites (for data see the apendix).

    The soil samples for this study were taken in 2014 from the upper 100 cm at five profiles at each the secondary rainforest with sawdust application (sawdust site) and the adjacent forest where no sawdust was applied (control site). The position of the ten profiles were randomly selected and had each a profile width of 1 m. We define horizon one and two as the topsoil and horizon three to five as the subsoil. Samples were oven dried at 40 °C in Ecuador at the Universidad Estatal Amazónica, before

transporting them to the German laboratory and sieving them to <2 mm.

    We classified the soil as an alusilandic Andosol, based on the WRB (2006) (for selected properties see Table 1). Few prominent X-ray diffraction reflexes indicate simple mineral composition. The crystalline primary minerals are amphibole, chlorite, quartz and plagioclase. Kaolinite and other secondary clay minerals are completely missing. Contents of crystalline

Fe minerals and gibbsite are little. Oxalate extractions indicate large amounts of short range-ordered and nano- or micro-crystalline mineral phases.





**Table 1.** Selected bulk properties of the studied Andosol. Soil layer thickness, pH values, bulk organic carbon concentration (OC) and carbon nitrogen ratios (C/N) are given as the mean and where convenient along with the standard error of data from five profiles. Bulk density (BD) is given as the mean of data from two profiles. $X_{ox}$ is the concentration of ammonium oxalate–oxalic acid extractable elements as the mean of data from three profiles. $Fe_d$ is the concentration of dithionite-extractable iron performed with samples from one profile. The concentrations of Al, Si and Fe are normalized to the mineral portion assuming that the mass of OM is two times the mass of OC (Sollins et al., 2006). Parenthesis represent standard deviation of sample parallels.

| horizon | layer thickness † | $pH_{CaCl_2}$ | BD † | OC † | C/N | $Al_{ox}$ | $Si_{ox}$ | $Al_{ox}/Si_{ox}$ | $Fe_{ox}$ | $Fe_d$ |
|---|---|---|---|---|---|---|---|---|---|---|
| | cm | - | $g\,cm^{-3}$ | $g\,kg^{-1}$ | - | $g\,kg^{-1}$ | $g\,kg^{-1}$ | molar ratio | $g\,kg^{-1}$ | $g\,kg^{-1}$ |
| control site | | | | | | | | | | |
| H1 | 10 (±1) | 4.1 | 0.26 | 212 (±16) | 12 | 55 (±2) | 12 (±2) | 4.7 (±0.8) | 21 (±3) | 20 |
| H2 | 13 (±5) | 4.6 | 0.38 | 128 (±9) | 11 | 67 (±7) | 23 (±8) | 3.1 (±0.7) | 20 (±2) | 20 |
| H3 | 24 (±4) | 4.8 | 0.38 | 75 (±0)* | 11 | 85 (±3) | 35 (±4) | 2.5 (±0.2) | 22 (±2) | 24 |
| H4 | 31 (±6) | 5.0 | 0.32 | 76 (±1) | 12 | 95 (±7) | 42 (±4) | 2.3 (±0.0) | 20 (±5) | 22 |
| H5 | 35 (±3) | 5.1 | 0.31 | 66 (±1) | 12 | 121 (±4) | 53 (±1) | 2.3 (±0.1) | 20 (±9) | 20 |
| sawdust site | | | | | | | | | | |
| H1 | 12 (±1) | 4.5 | 0.26 | 214 (±18) | 13 | 56 (±4) | 13 (±2) | 4.3 (±0.6) | 25 (±1) | 25 |
| H2 | 16 (±6) | 4.8 | 0.26 | 143 (±14) | 13 | 70 (±9) | 22 (±6) | 3.2 (±0.6) | 25 (±1) | 28 |
| H3 | 20 (±3) | 5.4 | 0.32 | 89 (±4)* | 12 | 76 (±6) | 29 (±5) | 2.7 (±0.3) | 28 (±2) | 26 |
| H4 | 29 (±3) | 5.6 | 0.38 | 77 (±4) | 12 | 84 (±20) | 35 (±9) | 2.4 (±0.1) | 28 (±7) | 24 |
| H5 | 28 (±9) | 5.6 | 0.32 | 67 (±3) | 12 | 92 (±13) | 39 (±3) | 2.4 (±0.2) | 31 (±8) | 33 |

† A Welch two sample t-test' was conducted with a p value of 0.05 for layer thickness, BD and bulk OC.

\* provides evidence that only the bulk OC concentrations in the third horizon (H3) differ significantly at the two sites.





## 2.2 Bulk organic carbon concentration and stock

Aliquots of all bulk samples were grounded, oven dried at 105 °C for 24 h prior to OC and nitrogen (N) determination with an Elementar Vario EL III CNS analyzer.

Organic carbon stocks were calculated based on soil volume to the fixed soil depth of 1 m. We found up to five soil layers per profile and determined bulk density, layer thickness and OC concentrations (Eq. (1)). Horizon thickness and OC concentrations were measured at all five profiles per site. The bulk density was only determined at two profiles per site (all horizons) and the mean (meanBD) was used for calculations. Thus OC stocks are presented as their mean and range, instead of standard derivation. As the soils contained only minimal amounts of material >2 mm, the soil particles < 2 mm represents the total soil

mass. For comparing OC stocks at different depths we also cumulated the OC stocks of each horizon proportionally. We choose the depth 0-25, 25-50 and 50-100 cm in order to represent the topsoil, horizon three and the subsoil below horizon three. We performed the 'Welch two sample t-test' of means (p = 0.05, non paired) for comparing bulk OC stocks.

$$OC\ stock\ [Mg\,ha^{-1}] = OC\ [g\,kg^{-1}] \cdot meanBD\ [kg\,dm^{-3}] \cdot layer\ thickness\ [dm] \tag{1}$$

    The equivalent soil mass approach propagated by Schrumpf et al. (2011); Wendt and Hauser (2013) was not applied, as i)

bulk density was not detected temporarily and did not vary much between sites for the same horizons, ii) the studied site was no cropland and iii) the approach increases uncertainties of OC stocks of undisturbed soils (Schrumpf et al., 2011).

## 2.3 Pyrogenic carbon analyses

Analysis of pyrogenic carbon (PyC) was peformed by the staff of the department of soil science at Rheinische Friedrich-Wilhelms-Universität Bonn. It was carried out following the revised protocol of Brodowski et al. (2005). For quantifying the

benzene polycarboxylic acids (BPCA), 10 mg of dried and ground soil material was treated with 10 ml 4 M $CF_3CO_2H$ (99%, Sigma Aldrich, Taufkirchen, Germany) to remove polyvalent cations. The PyC was then oxidized with $HNO_3$ (8 h, 170 °C) and converted to BPCAs. After cleanup via a cation exchange column (Dowex 50 W X 8, 200-400 mesh, Fluka, Steinheim, Germany), the BPCAs were silylated and determined using gas chromatography with flame ionization detection (GC-FID; Agilent 6890 gas chromatograph; Optima-5 column; 30 m × 0.25 mm i.d., 0.25 $\mu$m film thickness; Supelco, Steinheim, Germany). Two

internal standards citric acid and biphenyl dicarboxylic acid were used. Carefully monitoring the pH avoided decomposition of citric acid during sample processing as criticized by Schneider et al. (2010). The recovery of internal standard 1 (citric acid) ranged between 78 and 98 %. Carbon content of BPCA (BPCA-C) was converted to PyC with the conversion factor 2.3 (Brodowski et al., 2005). The analyses showed good repeatability, with differences between two measurement parallels being< 4.2 g PyC kg$^{-1}$OC except for the second horizon were the parallels differed by 12.5 g PyC kg$^{-1}$OC.





## 2.4 Sequential density fractionation of OM

We modified the sequential density fractionation procedure (Fig. 1) described by Cerli et al. (2012) in order to separate four different fractions. The first light fraction F1 contains mainly OM basically not interacting with the mineral phase often called free particulate OM. The second light fraction (F2) contains mainly particulate OM being incorporated into aggregates thus

having little interaction with the mineral phase. The third and forth fraction (F3, F4) are heavy fractions mainly containing OM strongly interacting with the mineral phase, which are often called organic-mineral associations.

Fifteen gram of dried (40 °C) and sieved (<2 mm) soil were mixed with 75 ml of sodium polytungstate solution (SPT, TC-Tungsten Compounds) with a density of $1.6 \, \mathrm{g \, cm^{-3}}$ in 200 ml PE bottles. To obtain F1, the bottles were gently shaken a

few times, and then the suspensions were allowed to settle for 1 h and subsequently centrifuged at $4500 \, g$ for 30 min (Sorvall RC-5B). The supernatant was siphoned with a water jet pump and the F1 fraction was collected on a pre-rinsed $1.2 \, \mu m$ cellulose-nitrate membrane filter. After rinsing with deionized water until the conductivity of the filtrate was $<50 \, \mu S \, cm^{-1}$, F1 was carefully transferred into a 50 ml PE bottle and subsequently freeze dried (Christ alpha 2-4 and 1-4 LCS).

The residue was re-suspended with re-collected SPT solution and refilled with fresh SPT solution ($1.6 \, \mathrm{g \, cm^{-3}}$) until the original bottle-sample mass was maintained. In order to release F2, the aggregates were dispersed by sonication (energy input $300 \, \mathrm{J \, ml^{-1}}$, with 10-mm pole head sonotrode, submersed to 15 mm depth, oscillation period 50/60 Hz, amplitude 2 AMP; Branson Sonifier 250) according to the calibration of Schmidt et al. (1999). The appropriate energy input was determined in a preliminary experiment as the energy which released the largest amount of largely pure OM, following Cerli et al. (2012).

Temperature was kept $<40 \, °C$ using an ice bath to avoid thermal sample alteration. Thereafter, the sample was centrifuge at $4500 \, g$ for 30 min and floating material was separated, washed and dried as described above for F1.

In order to further separate the residual fraction into Al-OM complexes and imogolite-type phases we introduced an additional density cut off. This is sensible because the overall density of an organic-mineral association depends on OM density,

mineral density and OM load (Kaiser and Guggenberger, 2007; Chenu and Plante, 2006). The densities of pure imogolite-type mineral phases and Al-OM complexes are very similar (Huang et al., 2011c), but Boudot (1992) and Kaiser and Guggenberger (2007) showed that Al-OM complexes have a higher OM load than imogolite-type phases. The second density cut off was set at $2.0 \, \mathrm{g \, cm^{-3}}$, which was determined in a preliminary experiment on the basis of OC concentrations, XRD spectra and oxalate-extractable Al, Si and Fe concentrations. The fraction with a density between 1.6. and $2.0 \, \mathrm{g \, cm^{-3}}$ was found to be enriched in

Al-OM complexes (F3).

For obtaining F3, the residue of the previous separation step was re-suspended in 75 ml fresh SPT solution (density of $2.0 \, \mathrm{g \, cm^{-3}}$), dispersed at $30 \, \mathrm{J \, ml^{-1}}$ to ensure complete soil wetting with new SPT solution, centrifuged, separated, washed and dried as described above for F1. The final residue of $>2.0 \, \mathrm{g \, cm^{-3}}$ density (F4), mainly containing imogolite-type phases and



crystalline minerals, was rinsed with deionized water until the conductivity of the supernatant was <50 $\mu$S cm$^{-1}$ and subsequently freeze-dried.

Aliquots of all fraction samples were oven dried at 105 °C for 24 h prior to OC and N determination with an Elementar Vario
5   EL III CNS analyzer.





**Table 2.** Selected bulk properties of samples used in the sequential density fractionation. Presented are bulk organic carbon (OC), carbon nitrogen ratios (C/N) and pyrogenic carbon (PyC) as proportion of OC. $X_{ox}$ is the concentration of ammonium oxalate–oxalic acid extractable elements. The concentrations of Al and Si are normalized to the mineral portion assuming that the mass of OM is two times the mass of OC (Sollins et al., 2006).

| horizon | depth | $pH_{CaCl_2}$ | OC | C/N | PyC | $Al_{ox}$ | $Si_{ox}$ | $Al_{ox}/Si_{ox}$ |
|---|---|---|---|---|---|---|---|---|
|  | cm | - | $g\,kg^{-1}$soil | - | $g\,kg^{-1}$OC | $g\,kg^{-1}$ | $g\,kg^{-1}$ | molar ratio |
| control site |  |  |  |  |  |  |  |  |
| H1 | 0-8 | 4.0 | 252 | 12 | 46 | 57 | 11 | 5.1 |
| H2 | 8-15 | 4.4 | 137 | 11 | 65 | 59 | 18 | 3.2 |
| H3 | 15-35 | 5.1 | 75 | 11 | 67 | 87 | 36 | 2.4 |
| H4 | 35-70 | 5.1 | 72 | 12 | 95 | 104 | 47 | 2.2 |
| H5 | 70-100 | 5.1 | 64 | 12 | 60 | 119 | 54 | 2.2 |
| sawdust site |  |  |  |  |  |  |  |  |
| H1 | 0-15 | 4.1 | 256 | 13 | 47 | 52 | 11 | 4.9 |
| H2 | 15-28 | 4.4 | 170 | 12 | 65 | 60 | 15 | 3.9 |
| H3 | 28-50 | 4.8 | 102 | 12 | 42 | 69 | 23 | 3.0 |
| H4 | 50-76 | 5.1 | 63 | 11 | 147 | 63 | 26 | 2.4 |
| H5 | 76-100 | 5.1 | 76 | 12 | 50 | 83 | 37 | 2.2 |





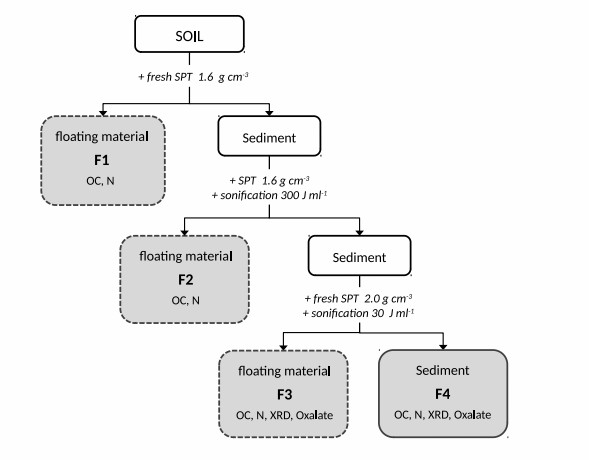

**Figure 1.** Sequential density fractionation scheme. SPT: sodium-polytungstate solution, F1: fraction which comprises material with densities <1.6 g cm$^{-3}$, in theory organic matter with basically no interaction with the mineral phase, F2: fraction which comprises material with densities <1.6 g cm$^{-3}$ and limited interaction with the mineral phase, F3: comprises material strongly interacting with mineral phases and overall soil particle densities between 1.6 and 2.0 g cm$^{-3}$. F4: comprises material strongly interacting with mineral phases and overall soil particle densities > 2.0 g cm$^{-3}$. Below the fraction labeling subsequent analysis are named. OC: total organic carbon concentration, N: nitrogen concentration, XRD: X-ray diffraction, Oxalate: ammonium oxalate–oxalic acid extraction

## 2.5 Acid oxalate extraction of F3 and F4

Aluminium (Al) and silicon (Si) of short range-ordered phases were extracted using the ammonium oxalate–oxalic acid reagent proposed by Schwertmann (1964). Either 0.1 or 0.5 g of oven dried (105 °C) and grounded F3 and F4 material, were suspended in 0.2 M ammonium oxalate–oxalic acid at pH 3 at a soil-to-solution ratio of 1 g:100 ml and shaken for two hours in the dark.

The suspension was decanted over MUNKEL 131 paper filters, with the first turbid efluent being discarded. Clear solutions were stored in the dark and at room temperature for no more than one day. Aluminium and silicon concentrations were determined by ICP-OES (Thermo Scientific iCAP 6000 series). The Fe concentrations were measured but results are not presented here since very low compared to other soils like Ferrasols (<30 g kg−1, Table 1). The recovery rates are calculated as the sum of the elements amount quantified in F3 and F4 normalized to bulk oxalate extracted amounts of the element. The results are

normalized to the mineral portion of the fraction assuming that the mass of OM is two times the mass of OC (Sollins et al., 2006).

## 2.6 X-ray diffraction of F3 and F4

X-ray diffraction (XRD) spectra were obtained on each one F3 and F4 sample per horizon of the samples from the sawdust site.

Samples were grounded with a ball mill and oven dried at 105 °C for 24 h. The random oriented powders were analyzed using

a PANalytical EMPYREAN X-ray diffractometer with theta/theta-geometry, 1 D-PIXcell detector, Cu-K$\alpha$ radiation at 40 kV and 40 mA, at an angle range of 5-65 °2$\theta$ with 378 s counting time per 0.013 ° 2$\theta$ step and automatically acting diaphragm. Evaluation was performed with X'Pert HighScore Plus V 3.0 (PANalytical) software.

## 2.7   Data analysis

All analyses, except for X-ray diffraction, which was carried out with no replicate, were carried out in duplicates. Bulk analyses are carried out with samples from five profiles per site. Sequential density fractionation and subsequent mineralogical analyses and PyC analyses were carried out for one representative profile per site (for selected soil data see Table 2). The soil profiles for these analyses were chosen on the basis of having five horizons within the upper 1 m, largest OC concentration in horizon one,

similar amount of acid oxalate-extractable elements and having different bulk OC concentrations in the third horizon. Results are presented as means of replicates. All calculations and graphs were processed with R version 3.1.0 (2014 The R Foundation for Statistical Computing).

## 3   Results

## 3.1   Bulk organic carbon

The total OC stocks are between 277 and 352 with a mean of 315 Mg ha$^{-1}$ at the sawdust site. At the control site total OC stocks range between 256 and 313 with a mean of 289 Mg ha$^{-1}$ at the control site (Table 3). The total OC stocks did not significant differe between the two sites. But out of the choosen depth sections the section of 25-50 cm, which comprises mostly the third horizon showed significant differences between the two sites (Table 3). In the third horizon the bulk OC concentrations

showed significantly differences, whereas the layer thickness and the bulk density did not (Table 1).

The PyC proportion on bulk OC range between 42 and 147 g PyC kg$^{-1}$OC (Table 2) and vary with depth in a non-regular pattern.





**Table 3.** Organic carbon (OC) stocks are presented as mean along with standard error (n = 5). † 'Welch two sample t-test' conducted with a
p value of 0.05 provides evidence that OC stocks differ significantly at the two sites only for the depth 25-50 cm.

| depth | OC stock † | |
|---|---|---|
| cm | Mg ha$^{-1}$ | |
| | control site | sawdust site |
| 0-25 | 111 ($\pm$7) | 116 ($\pm$8) |
| 25-50 | 66 ($\pm$2)* | 78 ($\pm$4)* |
| 50-100 | 113 ($\pm$5) | 120 ($\pm$5) |
| total | 289 ($\pm$9) | 315 ($\pm$15) |





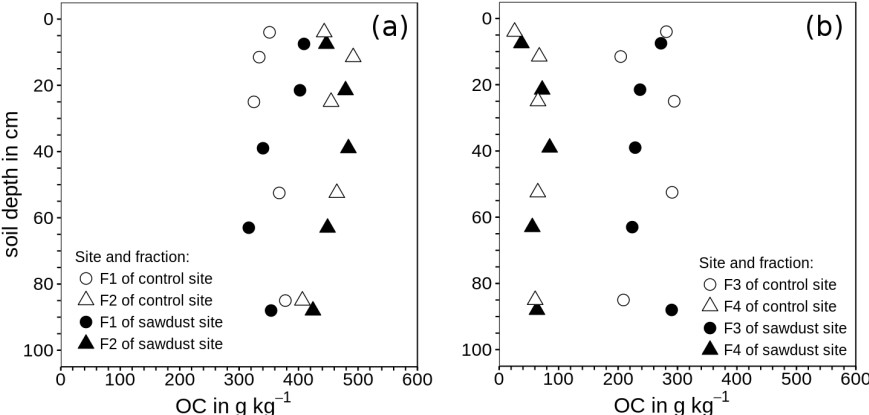

**Figure 2.** Mean of organic carbon (OC) concentration of density fractions one to four normalized to the mass of the respective fraction. Fractions 1 and 2 contain material (a), which was released directly (F1) and after applying $300\,\mathrm{J\,ml^{-1}}$ sonication energy (F2). Fraction 3 and 4 (b) comprise material strongly interacting with mineral phases and overall soil particle densities between 1.6-2.0 (F3) and $>2.0\,\mathrm{g\,cm^{-3}}$ (F4) respectively.

## 3.2 Sequential density fractionation

### 3.2.1 Organic matter analysis

The recoveries of the soil mass range between 98 and 102 wt%. The OC recoveries are on average 95 and always larger than 89 wt%.

5   The OC concentrations increase in the order F4 < F3 < F1 < F2 at all depths and all sites (Fig. 2). The OC concentrations are normalized to the respective fraction mass. The two light fractions (F1, F2) are rich in OC with 285-422 and 371-501 $\mathrm{g\,kg^{-1}}$. Fraction three shows also large OC concentrations, which vary slightly over depth between 204 and 294 $\mathrm{g\,kg^{-1}}$ changing in no particular pattern. In F4 the OC concentrations only increase within the first 10 cm from 30 to 65 $\mathrm{g\,kg^{-1}}$ and then remain constant for lower horizons.

  Fractions one and two account for less than 11 wt% of OC at the sawdust site and 20 wt% at the control site (Fig. 3, b). Their proportions decrease rapidly with depth to 2 and 1 wt% in horizon five. In general, the proportions of OC in F3 decrease drastically from > 65 wt% in the topsoil to <7 in the subsoil at both sites. The proportion of OC in F4 increase strongly from < 27 wt% in the topsoil to >91 in the subsoil at both sites.

  Figure 4, shows the OC concentrations of F3 and F4 normalized to bulk soil mass ($\mathrm{g\,kg^{-1}}$bulk soil). The F3 of the second horizon at the sawdust site contains 29 $\mathrm{g\,kg^{-1}}$soil more OC than F3 at the control site, which represents 91 wt% of the differences in bulk OC concentration. In the third horizon, F3 and F4 contain 11 and 14 $\mathrm{g\,kg^{-1}}$soil more OC than the respective




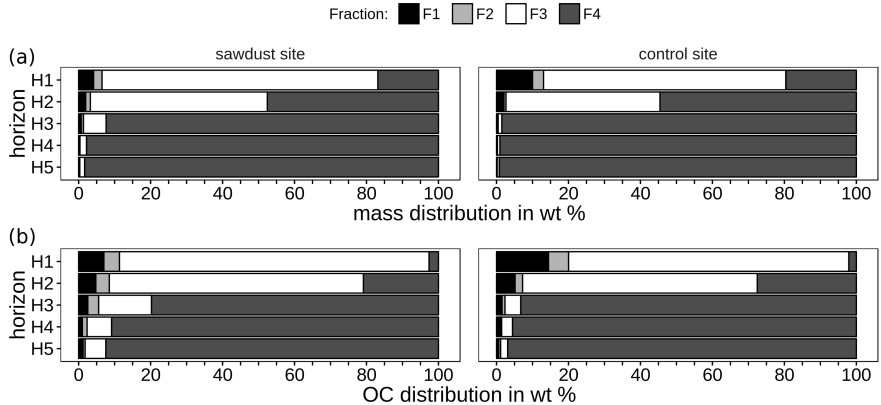

**Figure 3.** Mass (a) and organic carbon (OC, b) distribution in density fractions in percent of cumulated masses of fractions. Fraction 1 and 2 contain material, which was released directly (F1) and after applying $300 \, \mathrm{J \, ml^{-1}}$ sonication energy (F2). Fraction 3 and 4 comprise material strongly interacting with mineral phases and overall soil particle densities between 1.6-2.0 (F3) and $>2.0 \, \mathrm{g \, cm^{-3}}$ (F4) respectively.

fractions at the control site; combined they represent 93 wt% of the differences in the horizon's bulk OC concentrations. In the fourth and fifth horizon the OC concentration differences of the fractions between sites are $<3 \, \mathrm{g \, OC \, kg^{-1}}$ soil and not significant.

### 3.2.2 Acid oxalate extraction of F3 and F4

5 The recoveries of Al are on average 101 wt% and for Si systematicly 10 wt% lower, ranging between 87 and 96. The concentrations of oxalate-extractable Al are normalized to the mineral portion, assuming that the mass of OM is two times the mass of OC (Sollins et al., 2006). The concentrations of Al in F3 range between 63 and $227 \, \mathrm{g \, kg^{-1}}$ and are 2.1-6.7 times larger than those of F4, which range only between 5 and $128 \, \mathrm{g \, kg^{-1}}$ (Fig. 5). For both sites, the Al concentrations increase with increasing soil depth. The mineral portion-normalized Si concentrations show in general the same trend as the respective Al concentra-

10 tions for all fractions and sites with depth. They range between $11-61 \, \mathrm{g \, kg^{-1}}$ for F3 and $1-51 \, \mathrm{g \, kg^{-1}}$ for F4. The Al/Si molar ratios are always larger than two and are distributed in a concave function with depth, similar to bulk soil ratios (see Table 2 for data). Only the ratio in F4 of the uppermost horizon at the sawdust site is very low. At all sites, the Al/Si molar ratios in F3 are larger than the Al/Si molar ratios in F4 throughout the profiles.

### 3.2.3 X-ray diffractograms of F3 and F4

Differential X-ray spectra of F3 and F4 show similar main reflexes as the spectra of the bulk samples. The overall signal intensities in F4, are higher than for F3 spectra (Fig. 6). Moreover the F4 spectra show hardly any broad reflexions, while the F3 spectra have broad reflexions at 6-8 , 20-30 and $40 \, {}^\circ 2\theta$. These characteristics reflect low crystallinity material and are




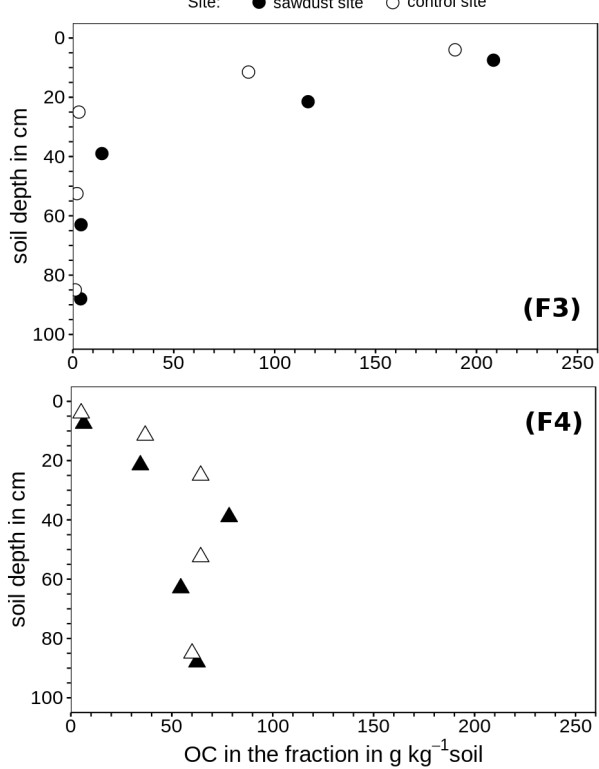

**Figure 4.** Organic carbon concentrations of fraction three and four normalized to the bulk soil mass. The fractions comprise material strongly interacting with mineral phases and overall soil particle densities between 1.6-2.0 (F3) and >2.0 g cm$^{-3}$ (F4) respectively.

assigned to imogolite-type mineral phases (Yoshinaga and Aomine, 1962; Basile-Doelsch et al., 2007). In contrast, the F4 spectra indicate larger amounts of crystalline minerals. Spectra of both fractions show broad reflexions at 6-8 and 20-30 $^{\circ}2\theta$ with increasing soil depth.

# 4 Discussion

## 4.1 Sequential density fractionation and oxalate extraction performance

Despite the numerous fractionation steps and extensive rinsing the overall high recoveries suggest very little losses of material due to dissolution and dispersion during the fractionation.

The OC concentrations in F2 (Fig. 2 a) are within the range of 400-500 g kg$^{-1}$, which Cerli et al. (2012) determined as almost pure organic matter. Thus the applied sonication energy of 300 J ml$^{-1}$ did not produce artefacts through redistribution



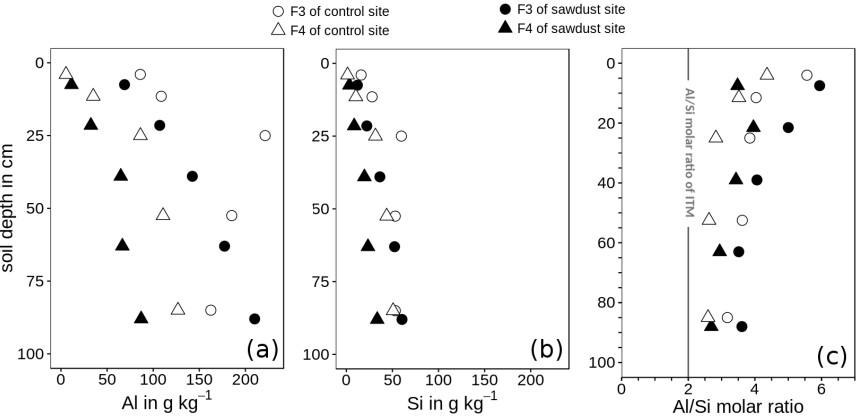

**Figure 5.** Concentration of acid oxalate-extractable aluminium (Al, a) and silica (Si, b) and Al/Si molar ratios (Al/Si, c) of the density fractions, which comprise material strongly interacting with mineral phases and overall soil particle densities between 1.6-2.0 (F3) and >2.0 g cm$^{-3}$ (F4), respectively. The metal concentrations are normalized to the mineral portion of the fraction, assuming that the mass of OM is two times the mass of OC (Sollins et al., 2006). ITM: imogolite-type mineral phases.

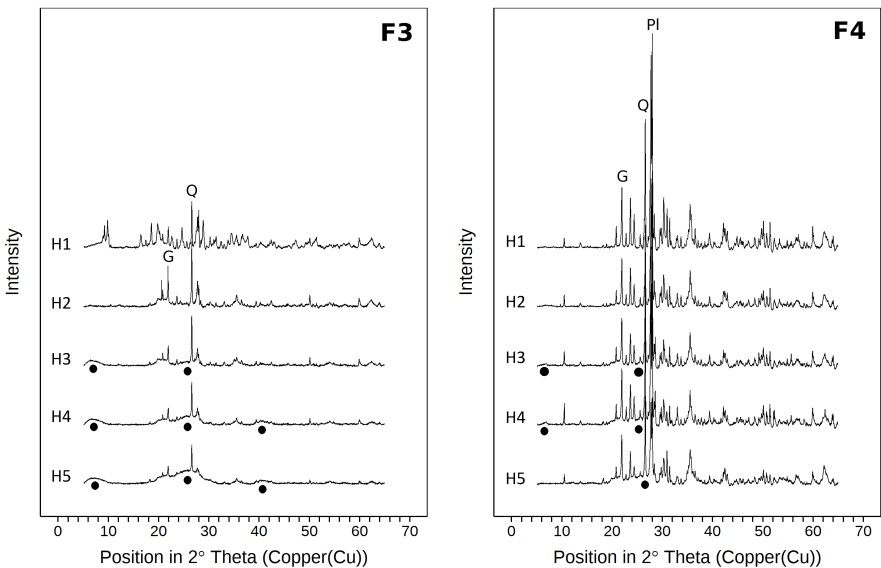

**Figure 6.** X-ray diffractograms (Cu-K$\alpha$ radiation) of the heavy fractions for all horizons of samples from the sawdust site. Fraction 3 and 4 comprise material strongly interacting with mineral phases and overall soil particle densities between 1.6-2.0 (F3) and >2.0 g cm$^{-3}$ (F4), respectively. All diffractograms are normalized to the same vertical scale. Circles indicate poorly crystalline material. Q marks the main quartz reflex. Pl marks the main plagioclase reflex. G marks the main gibbsite reflex.





of mineral phases over light fractions (Cerli et al., 2012). However, there is also no evidence for complete dispersion of aggregates, which Chenu and Plante (2006) state as impossible. The OC concentrations of F2 are larger than those of F1. This may be caused by sonication, which strips off basically all adhering mineral materials (Cerli et al., 2012). This „cleaning effect" leads to purer OM fractions in F2 than in F1. In consequence, the OC concentrations in F1 range only between 300-

400 g kg$^{-1}$. The weight proportions of OM ($x_{OM}$ as 2×OC) in F3 range between 0.4-0.6 g OM g$^{-1}$. We assume that the average mineral densities ($\rho_M$) in F3 are about 2.7 g cm$^{-3}$ (Huang et al., 2011c) and that OM is evenly distributed among the minerals with an average density ($\rho_{OM}$) of 1.4 g cm$^{-3}$ (Mayer et al., 2004). The resulting overall soil particle densities $\rho_{soil\,particle}$ calculated with Eq. (2) (Chenu and Plante, 2006) are between 1.7 and 2.0 g cm$^{-3}$, which is as expected from the density fractionation method. This indicates that the overall soil particle density in this Andosol is likely due to OM loadings

and not caused by variations in mineral density. This contrasts the results of Basile-Doelsch et al. (2007).

$$\rho_{soil\,particle} = \frac{a}{x_{OM}+b} \quad \text{with} \quad b = \frac{\rho_{OM}}{\rho_M - \rho_{OM}} \quad \text{and} \quad a = \rho_M \cdot b \tag{2}$$

The recoveries of acid oxalate-extractable Al and Si were large. The lower recovery of Si could be due to Si being present as silicic acid or silicon sorbed to ferrihydrite and other poorly crystalline phases (Childs, 1992), and may become desorbed during the sequential density fractionation.

## 4.2   Mineral composition of F3 and F4

Peak intensities of XRD spectra in F4 are up to 2.5 times larger than in F3. This indicates enrichment in crystalline minerals in F4 as compared to F3 and vice versa enrichment in short range-ordered phases in F3. The later enrichment is supported by the 2-7 times larger amount of oxalate-extractable Al in F3 than in F4. The broad signals in XRD spectra of F3 and F4 in deeper horizons suggest the presence of imogolite-type phases. The broad signals are less prominent for F4, because they are overlaid

by signals of crystalline minerals. On one hand, the Al/Si molar ratios of F3 are larger than the ratios of F4 in all horizons. Also, the C/Al molar ratios of F3 are larger than those of F4 throughout the profile, meaning that the organic-mineral associations in F3 have more OC. We conclude that F3 is enriched in Al-OM complexes compared to F4. On the other hand, we detected no differences in the assemblage of short range-ordered phases between fractions in the XRD spectra when comparing each horizon. Additionally, F4 of the topsoil has Al/Si molar ratios of 4, which means Al-OM complexes have to be present in ad-

dition to imogolite-type phases. Thus, a complete separation of Al-OM complexes and OM–loaded imogolite-type phases was not achieved by our density fractionation method. We think that Al-OM complexes and imogolite-type phases either form into continuous phases or that the density ranges of the two phases may overlap. Moreover, we think that quartz and other minerals could also be present in those fractions, because in the topsoil the XRD spectra of F3 show reflexes for primary minerals. This is also supported by the extremely low mineral portion-normalized Al and Si concentrations in F3 of the topsoil samples.

The Al/Si molar ratios decrease with depth, which indicates changes in the assemblage of short range-ordered phases. This allows for identifying those phases predominating the two heavy fractions of the different horizons. Many authors such as Yagasaki et al. (2006) use (Al-Al$_{py}$)/Si molar ratios, with Si and Al being oxalate-extractable Al and Si respectively, and





$Al_{py}$ being pyrophosphate-extractable Al. Pyrophosphate is supposed to extract Al from Al-OM complexes (Parfitt and Childs, 1988). We did not follow this approach, because the reliability of pyrophosphate–extraction has been questioned (Kaiser and Zech, 1996; Thompson et al., 2011). Kaiser and Zech (1996) attribute this to high pH of extractant, which can result in disso­lution of Al-containing mineral phases.

According to Amonette et al. (1994), oxalate-extractable Si originates almost exclusively from imogolite-type phases. The results of our oxalate–extraction show, that this Andosol is poor in Si concentration, and therefore, only imogolite-type phases with the minimum silicon content should be present. Instead of the pyrophosphate–extraction we proceeded as follows: We used the assumptions listed below and adapted the formula proposed by Parfitt et al. (1985) and Mizota and Reeuwijk (1989) with a fixed Al/Si molar ratio of 2 for Si-poor imogolite-type phases. With this it is possible to estimate imogolite-type phases, Al in Al-OM complexes and their molar portion in short range-ordered phases using the Eq. (3) to Eq. (5) (Table 4).

– oxalate-extractable Al (Al) is only incorporated in Al-OM complexes ($Al_{AOC}$) and imogolite-type phases ($Al_{ITP}$)

– Al-OM complexes comprise compounds which contain mainly Al-O-C bonds and scarcely any Al-O-Al bonds, because OM concentrations are high. Therefore we assume that Al-OM complexes contain on average 1.1 mol Al per 1 mol Al-OM complexes.

– oxalate-extractable Si is only incorporated in imogolite-type phases (= $Si_{ITP}$)

– $Al_{ITP}/Si_{ITP}$ molar ratio is 2

– $ITP_{cal}$ calculated concentration of imogolite-type phases

$$ITP_{cal} = 7.5 \cdot Si_{ITP} \tag{3}$$

$$Al_{AOC} = Al - 2 \cdot Si_{ITP} \tag{4}$$

$$ITP\,proportion = \frac{^{1}/_{2} \cdot Al_{ITP}}{^{1}/_{2} \cdot Al_{ITP} + ^{1}/_{1.1} \cdot Al_{AOC}} \cdot 100 \qquad \text{with Si and Al in } mol \cdot kg^{-1} \tag{5}$$





**Table 4.** Concentration of imogolite-type phase ($ITP_{cal}$), Al in Al-OM complexes ($Al_{AOC}$) and imogolite-type phase molar portions on the assemblage of short range-ordered phases (SRO) for F3 and F4, as calculated with Eq. (3) to Eq. (5). Data is presented as the mean and ranges in parenthesis.

| horizon | depth | $ITP_{cal}$ | | $Al_{AOC}$ | | molar portion ITP on SRO | |
|---|---|---|---|---|---|---|---|
| | cm | $g\,kg^{-1}$ | | $g\,kg^{-1}$ | | mol % | |
| | | F3 | F4 | F3 | F4 | F3 | F4 |
| control site | | | | | | | |
| H1 | 4 | 122 ($\pm$2) | 10 ($\pm$1) | 55 ($\pm$0) | 3 ($\pm$0) | 24 | 32 |
| H2 | 12 | 212 ($\pm$23) | 78 ($\pm$0) | 55 ($\pm$6) | 15 ($\pm$1) | 35 | 42 |
| H3 | 25 | 452 ($\pm$11) | 239 ($\pm$3) | 106 ($\pm$3) | 25 ($\pm$0) | 37 | 57 |
| H4 | 52 | 402 ($\pm$0) | 332 ($\pm$4) | 83 ($\pm$0) | 26 ($\pm$0) | 40 | 63 |
| H5 | 85 | 402 ($\pm$0) | 384 ($\pm$3) | 60 ($\pm$0) | 29 ($\pm$0) | 48 | 65 |
| sawdust site | | | | | | | |
| H1 | 8 | 91 ($\pm$5) | 26 ($\pm$4) | 46 ($\pm$5) | 5 ($\pm$1) | 22 | 42 |
| H2 | 22 | 168 ($\pm$3) | 64 ($\pm$3) | 64 ($\pm$1) | 16 ($\pm$1) | 27 | 36 |
| H3 | 39 | 276 ($\pm$2) | 149 ($\pm$0) | 72 ($\pm$1) | 27 ($\pm$1) | 35 | 43 |
| H4 | 63 | 396 ($\pm$5) | 178 ($\pm$0) | 77 ($\pm$4) | 21 ($\pm$1) | 42 | 54 |
| H5 | 88 | 457 ($\pm$4) | 254 ($\pm$2) | 94 ($\pm$2) | 22 ($\pm$1) | 40 | 61 |



We estimated the weight proportion of short range-ordered phases from F3 and F4 on the bulk mineral mass and divided them into the groups "very abundant", "abundant", "low" and "traces" by using Al and Si mass distributions in the different fractions (Table 5). Results show that in the topsoil short range-ordered phases are mostly present in F3 and in the subsoil they are mostly present in F4. On the basis of the calculated imogolite-type phase molar portions we decided on the prevalent

species of the short range-ordered phase assemblages (Table 5). In the topsoil, Al-OM complexes prevail, because imogolite-type phases contribute only with $< 30\,\mathrm{wt\%}$ to short range-ordered phases of F3. In the subsoil, the presence of imogolite-type phases and Al-OM complexes is more balanced in F4, with increasing portions of imogolite-type phases with depth.

The characteristic broad signals indicating imogolite-type phases in the X-ray diffractograms appear in the subsoil, but not

in the topsoil (Fig. 6). Imogolite-type phases dissolve at pH values below 4.8 (Huang et al., 2011b). The pH values in the subsoil are equal or above 4.8, whereas pH values in the topsoil are below 4.8 (Table 1). Thus, the X-ray diffractograms and pH values support our suggestion and explain the presence of imogolite-type phases in the subsoil and their absence in the topsoil. We conclude that the studied Andosol shows aluandic properties in the topsoil and silandic properties in the subsoil.

## 4.3  Organic carbon storage forms

In comparison to Amazonian Oxisols (1-3 g PyC kg$^{-1}$ soil, 140 g PyC kg$^{-1}$ OC, Glaser et al. (2000)) and Terra Preta soils (4-20 g PyC kg$^{-1}$ soil, 350 g PyC kg$^{-1}$ OC, Glaser et al. (2000)), the studied Andosol has medium to high PyC concentrations. However, PyC plays only a marginal role for the accumulation of total OM because it contributes only up to 5 wt% to the OC concentration. We consider the later conclusion true, although the conversion factor of 2.3 is criticized by Schneider et al.

(2010) as inaccurate.

Only up to 20 wt% of OM in the topsoil and 2 wt% in the subsoil is not bound to mineral phases. The low proportions of OC in the light fractions is in line with data published by Paul et al. (2008) for topsoils of Andosols under tropical rainforest (20 wt% for material with densities $<1.6\,\mathrm{g\,cm^{-3}}$). This is caused by rapid mineralization of particulate OM and the strong

interaction of minerals with OM in Andosols (Basile-Doelsch et al., 2007; Huang et al., 2011a). Thus, the mineral phase plays the dominant role in stabilizing OM in this Andosol.

We used OC mass distributions in the different fractions for dividing OC from F3 and F4 into the groups "very abundant", "abundant", "low" and "traces" (Table 5). The OC abundance clearly correlates with the distribution of mineral phases in both

fractions. In the topsoil, OC is mainly associated with Al-OM complexes, whereas the OC in the subsoil is mainly bonded to imogolite-type phases. Based on the results in section 4.2 we suggest that Al-OM complexes are in close contact to imogolite-type phases e.g. precipitate on the surfaces.





**Table 5.** Mineralogical control of organic carbon (OC) stabilization in fraction three and four. The mass proportions in the different fractions for OC and short range-ordered and nano- or micro-crystalline phases (SRO) range from "+++" (very abundant, > 75 wt%), "++" (abundant, 75-30 wt%), "+" (low, 30-3 wt%) to "tr" (traces, < 3 wt%). The prevalent short range-ordered species are defined by the imogolite-type phase molar proportion on the short range-ordered phase (Table 4). For the mean molar proportions of both sites < 33 wt% Al-OM complexes prevail. For molar proportions between 33 and 67 wt% Al-OM complexes and imogolite-type phases are balanced, with the first mentioned short range-ordered species being present in slightly larger portion than the other. For molar proportions > 67 wt% imogolite-type phases prevail. ITP: imogolite-type phases. AOC: Al-OM complexes

| horizon | F3 | | | F4 | | |
|---------|---------------|----------------|----------------------|---------------|----------------|----------------------|
| | OC proportion | SRO proportion | prevalent SRO species | OM proportion | SRO proportion | prevalent SRO species |
| H1 | +++ | +++ | AOC | tr | tr | AOC, ITP |
| H2 | ++ | ++ | AOC | + | + | AOC, ITP |
| H3 | + | + | AOC, ITP | +++ | +++ | AOC, ITP |
| H4 | tr | tr | AOC, ITP | +++ | +++ | ITP, AOC |
| H5 | tr | tr | AOC, ITP | +++ | +++ | ITP, AOC |





## 4.4 Organic carbon response to sawdust input

No sawdust was optically visible neither in the soil profile nor in the light density fractions prior to grounding. Chambers et al. (2000) report that wood density and bole diameter were significantly and inversely correlated with the decomposition rate constants for dead trees in tropical forests of the central Amazon. For the smallest bole diameter (10 cm) the authors calculated

$0.26\,\mathrm{year}^{-1}$ as the lowest rate constant. A dead tree with such a diameter would then be decomposed to 99 % after 17 years. Additionally Powers et al. (2009) showed that in tropical soils the decomposition rate increases lineary with the precipitation. With sawdust beeing much finer in texture than a dead tree and the precipitation at the CERFA site being twice as high as at the site of Chambers et al. (2000), we expect a much faster decomposition than 17 years. Moreover, the phosphorus concentrations, determined in an aqua regia solution for samples taken from the first 20 cm (data not shown) are significantly larger

at the sawdust site than at the control site. The additional phosphorus at the sawdust site matches the phosphorus input via sawdust. Additionally, the C/N ratios of all fractions in the upper two horizons are below 20, showing no difference between the two sites. From all this we conclude that the added sawdust, which has a C/N mass ratio of around 110, was completely decomposed on-site.

The sawdust did not significantly increase the bulk OC concentrations in the upper two horizons. We also found no indications of additional inclusion of OM into macro-aggregates. This holds true despite the fact that the sample of the second horizon used for the sequential density fractionation at the sawdust site has a larger OC concentration than the sample at the control site. Those results have to be interpreted with some caution in terms of site comparison, because we conducted the fractionation experiment only with one profile per site. The sequential density fractionation also revealed that over 80-89 wt%

of OC are strongly associated with minerals. We therefore conclude that the topsoil is saturated with OC and not even small parts of the massive carbon input can be additionally stored. We assign this to limited and saturated storage capacities of the mineral phase.

Kaiser and Kalbitz (2012) state that in soils, where percolating water controls transport processes a steady input of surface-

reactive plant derived compounds force less strongly binding compounds to move further down. Thus, OM migrates downwards in form of dissolved OM if the OM storage capacities in the top layer are exhausted. When reaching soil horizons with free storage capacity the dissolved OM compounds are retained and OC concentration increases. This would explain the significantly larger OC concentration in the third horizon at the sawdust site. As long as the third horizon has free storage capacities, the OC concentration in horizon four and five will not increase, which is in line with our results. Over 90 wt% of the additional

OM in the third horizon are recovered in F3 and F4 together. Thus, the increase in bulk OC concentration in the third horizon is due to OM strongly interacting with the mineral phase and thus of long term stability.

The increase in OC concentration in the third horizon in response to the added sawdust could, with respect to the mechanisms, be either due to undersaturated sorption complexes or a change in carbon storage mechanisms towards mechanisms



with higher carbon storage capacities. It is noticeable, that the larger OC bulk concentration in the sample of the third horizon used in the fractionation experiment of the sawdust site correlates with a larger proportion of F3 and slightly lower pH values compared to the third horizon at the control site. The lower pH values are not the result of the mineralization of $18\,\mathrm{Mg\,ha^{-1}}$ nitrogen, which were added along with the sawdust, because the decomposition of sawdust was complete and therefore basic

cations were also released. Those basic cations buffer the released protons from the nitrification reaction (Breemen et al., 1983). This result suggest that the percolating DOM solution also influences the short range-ordered phase composition via reactions with protons. This implies that previously formed imogolite-type phases may be in situ transformed into Al-OM complexes in response to changes in pH and OM input.

The OC stock increased significantly at the sawdust site for the depth section of 25-50 cm, which belongs to the subsoil and comprises mostly the third horizon. We found that this difference was due to the increase in OC concentration and not caused by neither layer thickness, nor bulk density variations. Despite this increased in OC stock in parts of the subsoil, we found no evidence that the added sawdust increased the total OC stock significantly. We think that this is not due to smaller numbers of bulk density measurements compared to all other parameters, because Schrumpf et al. (2011) found that the contribution of

bulk density to the OC stock variability was lower and decreased with soil depth than the contribution of OC concentration and layer thickness. We rather think, that the large variability of OC concentrations in the topsoil overlay the effect of the larger OC concentration in the subsoil at the sawdust site. To evaluate the OC accumulation, we calculated the increase in OC stock within the depth section 25-50 cm (Tab 3). The resulting additional OC stock at the sawdust sites is $12\,\mathrm{Mg\,ha^{-1}}$, which represents only 0.7 wt% of the originally added $1800\,\mathrm{Mg\,OC\,ha^{-1}}$. Thus, the studied Andosol has an extremely low OC accumulation

rate. Six et al. (2002) postulate in their saturation concept that soils, which are close to their maximum OC storage capacity have low accumulation rates.

## 5 Conclusions

The massive OC input did not increase the OC concentration in topsoils but in the subsoil, which resulted in significantly larger OC stocks for the subsoil. Thus, the OC-rich Andosol topsoils are not capable of storing additional carbon, likely because of

limited binding capacities of their mineral phases. Seemingly, some of the additional OC migrates downwards with the percolating water until reaching layers where free binding sites are available. The studied soils, hence, are saturated with OC when only considering the topsoils but still have large capacity to host more OC in their deeper layers.

The OC increase in the subsoil was exclusively due to binding to mineral phases. Since binding to mineral phases promotes

retarded mineralization, i.e., longer turnover times, stabilization, and thus, long-term storage of the additional OC can be expected.





The additional OC was likely stored within Al-OM complexes and by binding to imogolite-type phases. There are indications that the input of additional OC into the subsoils induced dissolution of the imogolite-type mineral phases and subsequent formation of Al-OM complexes. This transition from a predominately silandic to a more aluandic mineral assemblage would increase the subsoils storage capacity for OC. It also suggest that silandic Andosols can gradually become aluandic.

5     Despite the increase in subsoil OC, there was no significant change in total OC stocks in response to the massive OC inputs over a period of 20 years. This was basically because of spatial variations that demand for larger changes than the observed ones to become significantly different.

    The results clearly show that accumulation efficiency of the added OC was very low. Increasing the OC stock in soils already
10   rich in OC requires comparably large inputs over long time periods to induce OC transport into the deeper soil layers. This contrasts the situation in young soils where OC stocks build up rapidly in surface-near layers.



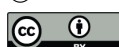

**Table 1.** gravimetric determined rooting intensity, measured at 5 soil cores at each site

| depth | mean | standard deviation | standard error |
|---|---|---|---|
| cm | Mg ha$^{-1}$ | Mg ha$^{-1}$ | Mg ha$^{-1}$ |
| control site | | | |
| 0-20 | 3.8 | 1.3 | 0.3 |
| 20-40 | 0.7 | 0.7 | 0.1 |
| sawdust site | | | |
| 0-20 | 5.5 | 2.6 | 0.5 |
| 20-40 | 0.4 | 0.3 | 0.1 |



*Competing interests.* The authors declare that they have no conflict of interest.

5  *Acknowledgements.* The authors acknowledge the financial support of the *Technische Universität Berlin*. The pyrogenic carbon analysis were conducted by Arne Kappenberg at the department of soil science at the *Rheinische Friedrich-Wilhelms-Universität Bonn*, for which we are grateful. We also acknowledge Nelson Omar Tello Benalcázar and Josue Tenorio for providing the study sites and assisting during the field work.



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
