# Peer review of "Massive carbon addition to an organic-rich Andosol increased the subsoil but not the topsoil carbon stock."

_Biogeosciences, 2017_

## Referee Comment (RC1) · Anonymous Referee #2 · 27 Nov 2017

The authors have revised the manuscript very well. I have only some minor comments. 1) In the Abstract, please revise it in a concise way. For example, the main result is in the line 15, only a sentence. 2) Please give the full name of 'WRB' in page 4 Line 22. 3) In the Table 1, please give full information of 'Xox' in the table title in Page 5. Second, what is the unit for Alox and other mineral, g kg-1 dry soil? or g kg-1 organic C? 4) The similar description as abovementioned for Table 2 in Page 9. 5) In the result section, '3.2 sequential density fractionation' in Page 13 and Page 14, the description is not clear, please revise it in a concise way. 6) In the figure 2, I am wondering why the sampling depths at the two sites are different? 7) In Page 18, the methods and calculation description need to put in the material section? I am wondering why.

---

## Referee Comment (RC2) · Anonymous Referee #3 · 29 Nov 2017

The manuscript brings interesting conclusions about the carbon storage capacity of andosols. The analyzes are well documented and lead to the description of interesting mechanisms to explain an important carbon storage capacity in the subsoil when the topsoil binding capacities are exhausted. The manuscript would need a thorough English proofreading. Comments and suggestions: 1) The authors could reformulate the title to state the positive result first (C storage in subsoils). 2) Page 2, L.3: "Soil holds more organic carbon (OC) than there is carbon in the global vegetation and atmosphere combined": the authors should provide reference(s) from scientific literature. 3) Page 2, L. 12: "the main factor": the authors should be more specific (of what?). 4) Page 4, L. 9-10: the authors could discuss the fact that higher vegetation in the sawdust site may have consequences on carbon intakes from the soil. Could the fact that vegetation is higher be due to sawdust additions? 5) Page 4, L. 16-19: a schema or photo of a soil profile would help to understand how horizons were delimited. It could be added as an appendix. 6) Page 5, Table 1: For each element presented in the table, the number of data that were pooled to calculate the mean value could be specified in the table (eg : "BD, g.cm-3, n=2" or "C/N, n=5). Standard deviations (rather than standard errors) should probably be used in tables. In column BD, why isn't the standard error/deviation specified? 7) Page 9, Table 2 / Page 11, L.7: How was the "representative" profile chosen? 8) Page 13, L. 3 / Page 14, L.5: The authors should explain how values can exceed 100 wt%. 9) Page 13, L. 5-6 / Page 14, L.6: Could the authors explain why normalizations were performed? 10) Paragraph 4.4: The authors should emphasize more on the fact that the very low number of data used for comparing the sites may also be responsible for the absence of significative differences. 11) Page 23, L.30: The authors could add a sentence on the potential of mineral phases for carbon storage. 12) Page 24, L. 3-5: the absence of significative difference could also be due to the fact that the number of data was to small to detect it.

---

## Referee Comment (RC3) · Anonymous Referee #4 · 3 Dec 2017

The manuscript answers the specific question of how much organic carbon (OC) can be stored in a specific type of soil (Andosol), giving experimental support to the theory of a finite OC accumulation potential of soils due to the limited binding capacities of minerals. I found the manuscript well written and well organized. Even though the experimental site was not designed from the beginning for this trial, I agree with the authors saying that it is rare to have such a long history of treatments, and this is particularly valuable when studying carbon dynamics in soil. My bigger concern regards the bulk density (BD) values reported in table 1 and the way they were assessed. As reported in equation 1, BD is of a pivotal importance in assessing the total amount of OC in the soil (Mg/ha), so I think it should be specified with more details how the sampling was carried out, which volume of soil was considered, which corer and so on. Ten sampling points don't represent a huge number, but for BD just 2 were used, which is really low for such a big area. This observation arises because the BD values reported in table one are really low even for andosols (< 0.4 Mg/m3), which causes a relatively low amount of OC per hectare considering the relevant percentage that OC reaches in that soil (up to 21%). For instance, 315 Mg OC/ha found in this study (considering 1m soil depth) are obtainable with a 2.6% of OC in a soil with BD = 1.2 Mg/m3, very typical in mineral soils. Thus, besides improving the description of the sampling methodology, I suggest adding some reference supporting such low BD values. I would also stress this fact more in the discussion chapter, section 4.4: the low BD is likely due to a low content of minerals, which can be "easily" saturated with OC at least in the upper layers. Please provide also some more info about the way in which sawdust was distributed over the years, specifying to which degree can the soil treated with sawdust considered homogeneous within the 3 hectares of the trial. Some technical notes: P1 – L3: please add a reference at the end of the first sentence (storage of organic carbon in soils) P4 - L11: please use the same decimals (1-16 o 0.9 – 16.0 Mg ha-1 year-1) P6 – L9: standard deviation (not derivation)

---

## Author Comment (AC1) · 20 Dec 2017

Dear reviewer #2, we thank you for taking once again the time to provide us with a second feedback on the manuscript. We carefully considered your comments. Our responses and suggestions for possible changes are given below each comment. We will upload the revised manuscript at the end of the discussion together with changes suggested by other reviewers.
The authors have revised the manuscript very well. I have only some minor comments.

1) In the Abstract, please revise it in a concise way. For example, the main result is in the line 15, only a sentence.

Author response: We will summarise the main result in one sentence at the end of the abstract.

2) Please give the full name of 'WRB' in page 4 Line22.

Author response: We will provide the full soil names according to the World Reference Base for Soil Resources (2015) in the Materials section.

3) In the Table 1, please give full information of 'Xox' in the table title in Page 5. Second, what is the unit for Alox and other mineral, g kg-1 dry soil? or g kg-1 organic C?

Author response: We will give full reference to Alox, Siox and Feox in the table title instead of referring to Xox. The oxalate-extractable metals are presented as g kg-1 mineral part (or inorganic part) of the dry soil. We used this unit in order to evaluate the amount of oxalate-extractable metals in relation to total mineral constituents. The large and strongly varying concentrations of organic matter with depth masks the actual proportion of oxalate extractable minerals. For better comparability, we normalised the oxalate-extractable metals to the mineral soil component instead of to dry soil. For clarification, we will add explanations to section "2.5. Acid oxalate extraction..." and the table titles.

4) The similar description as above mentioned for Table 2 in Page 9.

Author response: see above (3)

5) In the result section, '3.2 sequential density fractionation' in Page 13 and Page 14, the description is not clear, please revise it in a concise way.

Author response: We assume that you find section 3.2.1 (p. 13 line 3 to p. 14 line 2) not clear. We will add additional information on the purpose of the data for each of the

three paragraphs. However, we are not certain, what is unclear. Could you be so kind to give us more detailed suggestions? That would be very helpful.

6) In the figure 2, I am wondering why the sampling depths at the two sites are different?

Author response: We sampled in the middle of each horizon. Horizon thickness varied slightly between profiles, which is a common feature of forest soils. In result also the sampling depths for the profiles selected for density fractionation were not identical.

7) In Page 18, the methods and calculation description need to put in the material section? I am wondering why.

Author response: We think that this is a good point. We also discussed several times where to put the calculation description. As the hypothetical arguments became relevant after we obtained the results of the oxalate extraction and because their are disputable we choose to put them in the discussion section. Nevertheless they could be just as well be illustrated in the methods section.

---

## Author Comment (AC2) · 16 Jan 2018

Dear reviewer #3, we thank you for taking the time to provide us with your feedback on our manuscript. We carefully considered your comments. Our responses and suggestions for possible changes are given below each comment. We upload the revised manuscript at the end of the discussion together with changes suggested by other reviewers.

The manuscript brings interesting conclusions about the carbon storage capacity of

andosols. The analyzes are well documented and lead to the description of interesting mechanisms to explain an important carbon storage capacity in the subsoil when the topsoil binding capacities are exhausted.

The manuscript would need a thorough English proofreading.

Author response: A professional English proofreading will be carried out.

Comments and suggestions:

1) The authors could reformulate the title to state the positive result first (C storage in subsoils).

Author response: We change the title to "Massive carbon addition to an organic-rich Andosol increased the subsoil but not the topsoil carbon stock."

2) Page 2, L.3: "Soil holds more organic carbon (OC) than there is carbon in the global vegetation and atmosphere combined": the authors should provide reference(s) from scientific literature.

Author response: We provide the reference: Lehmann, Johannes, and Markus Kleber. "The Contentious Nature of Soil Organic Matter." Nature 528, no. 7580 (Dezember 2015): 60–68. https://doi.org/10.1038/nature16069.

3) Page 2, L. 12: "the main factor": the authors should be more specific (of what?).

Author response: We change the sentence to: "Paustian et al. (1997) consider the OC stocks to increase linearly and limitless with increasing organic input."

4)Page 4, L. 9-10: the authors could discuss the fact that higher vegetation in the sawdust site may have consequences on carbon intakes from the soil. Could the fact that vegetation is higher be due to sawdust additions?

Author response: Unfortunately, we have no data of possible differences in plant biomass and plant community composition. We calculated a worst case scenario to exclude possible differences in C input with litter between the sites. The scenario shows, that differences in input of plant-derived C between sites are insignificant relative to the sawdust input. We clarify the purpose of the scenario calculation in the manuscript.

5) Page 4, L. 16-19: a schema or photo of a soil profile would help to understand how horizons were delimited. It could be added as an appendix.

Author response: We provide pictures of the soil profiles in the appendix or if suitable in the "materials and methods" section.

6) Page 5, Table 1: For each element presented in the table, the number of data that were pooled to calculate the mean value could be specified in the table (eg : "BD, g.cm-3, n=2" or "C/N, n=5). Standard deviations (rather than standard errors) should probably be used in tables. In column BD, why isn't the standard error/deviation specified?

Author response: We add a row in the table showing the number of data that were pooled. We add the standard error of BD. According to Altman and Bland (2005) and Reinhart (2015), the standard deviation (SD) is a measure of variability and describes the spread of individual data points. The standard error (SE = SD/sqrt(n)) is a type of standard derivation, which describes how far the average for this sample might be from the true average. As we are more interested in the latter, we choose the standard error.

Altman, Douglas G, and J Martin Bland. "Standard Deviations and Standard Errors." BMJ : British Medical Journal 331, no. 7521 (October 15, 2005): 903.

Reinhart, Alex. Statistics Done Wrong: The Woefully Complete Guide. No Starch Press, 2015.

7) Page 9, Table 2 / Page 11, L.7: How was the "representative" profile chosen?

Author response: The representative soil profiles were selected to meet the following criteria: five horizons within the upper 1 m, largest OC concentration in the topmost horizon, similar amounts of acid oxalate-extractable Al, Fe, and Si, and having different

bulk OC concentrations in the third horizon. This is already lined out in section "2.7 Data analysis". We add the decision basis to the caption of Table 2.

8) Page 13, L. 3 / Page 14, L.5: The authors should explain how values can exceed 100 wt%.

Author response: We add following sentence to section 4.1.: "Values exceeding 100wt% are probably caused by random errors of measurements and some sodium polytungstate not entirely removed by sample washing."

9) Page 13, L. 5-6 / Page 14, L.6: Could the authors explain why normalizations were performed?

Author response: We change the sentence to: "The OC concentrations are related to the respective fraction mass."

10) Paragraph 4.4: The authors should emphasize more on the fact that the very low number of data used for comparing the sites may also be responsible for the absence of significative differences.

Author response: We agree with you and include the calculation of statistical power and its results to the manuscript

11) Page 23,L.30: The authors could add a sentence on the potential of mineral phases for carbon storage.

Author response: We add a sentence in the conclusion accordingly.

12) Page 24, L. 5-7: the absence of significative difference could also be due to the fact that the number of data was to small to detect it.

Author response: We emphasize this accordingly.
* * *

---

## Author Comment (AC3) · 16 Jan 2018

Dear reviewer 4,

we thank you for taking the time to provide us with your feedback on our manuscript. We carefully considered your comments. Our responses and suggestions for possible changes are given below each comment. We upload the revised manuscript at the end of the discussion together with changes suggested by other reviewers.

The manuscript answers the specific question of how much organic carbon (OC) can

be stored in a specific type of soil (Andosol), giving experimental support to the theory of a finite OC accumulation potential of soils due to the limited binding capacities of minerals. I found the manuscript well written and well organized. Even though the experimental site was not designed from the beginning for this trial, I agree with the authors saying that it is rare to have such a long history of treatments, and this is particularly valuable when studying carbon dynamics in soil.

My bigger concern regards the bulk density (BD) values reported in table 1 and the way they were assessed. As reported in equation 1, BD is of a pivotal importance in assessing the total amount of OC in the soil (Mg/ha), so I think it should be specified with more details how the sampling was carried out, which volume of soil was considered, which corer and so on. Ten sampling points don't represent a huge number, but for BD just 2 were used, which is really low for such a big area.

Author response: We agree that the BD is of pivotal importance. We add the standard error in Table 1. The bulk density was determined at two profiles per site (all horizons). For each horizon, five replicates were sampled with 100 cm$^3$ corers, oven-dried at 105°C for 24 hours and weighted. We think that the number of bulk density measurements are sufficient, because the standard error was very small (with <0.01 g cm$^{-3}$). We add those details in the revised manuscript.

This observation arises because the BD values reported in table one are really low even for andosols (< 0.4 Mg/m3), which causes a relatively low amount of OC per hectare considering the relevant percentage that OC reaches in that soil (up to 21

Author response: We add some references supporting BD values below 0.5 g cm$^{-3}$ in Andosols. Especially aluandic Andosols feature often such low BD values. We also add some sentences to Section 4.4 to compare the OC stock of this Andosol with values from other soils.

Basile-Doelsch, I., R. Amundson, W.E.E. Stone, D. Borschneck, J. Y. Bottero, S. Moustier, F. Masin, and F. Colin. "Mineral Control of Carbon Pools in a Volcanic Soil Horizon." Geoderma 137, no. 3–4 (2007): 477–489. https://doi.org/10.1016/j.geoderma.2006.10.006.

Marin-Spiotta, Erika, Oliver A. Chadwick, Marc Kramer, and Mariah S. Carbone. "Carbon Delivery to Deep Mineral Horizons in Hawaiian Rain Forest Soils." Journal of Geophysical Research: Biogeosciences 116, no. G3 (2011): n/a–n/a. https://doi.org/10.1029/2010JG001587.

Tonneijck, Femke H., B. Jansen, K. G. J. Nierop, Jacobus M. Verstraten, Jan Sevink, and L. de Lange. "Towards Understanding of Carbon Stocks and Stabilization in Volcanic Ash Soils in Natural Andean Ecosystems of Northern Ecuador." European Journal of Soil Science 61, no. 3 (2010): 392–405. https://doi.org/10.1111/j.1365-2389.2010.01241.x.

I would also stress this fact more in the discussion chapter, section 4.4: the low BD is likely due to a low content of minerals, which can be "easily" saturated with OC at least in the upper layers. Please provide also some more info about the way in which sawdust was distributed over the years, specifying to which degree can the soil treated with sawdust considered homogeneous within the 3 hectares of the trial.

Author response: Yes, the smaller content of mineral phases in the topsoil might cause the smaller sorption capacity but probably the smaller mineral content itself is just a result of the strong sorption of organic matter by highly reactive mineral phases. The sawdust was distributed approximately evenly over the site by hand. We add this information to the Materials and Methods section.

Some technical notes:

P1– L3: please add a reference at the end of the first sentence (storage of organic carbon in soils)

P4 - L11: please use the same decimals (1-16 o 0.9 – 16.0 Mg ha-1 year-1)

P6 – L9: standard deviation (not derivation)

Author response: Thank you for pointing this out. We change the manuscript accordingly.

––––––––––––––––––––––––––––––––

---

## Author Response (AR1)

**Massive carbon addition to an organic-rich Andosol increased the subsoil but not the topsoil carbon stock.**

Antonia Zieger[1], Klaus Kaiser[2], Pedro Ríos Guayasamín[3], and Martin Kaupenjohann[1]

[1]Department of Soil Science, Institute of Ecology, Technische Universität Berlin, Ernst-Reuter-Platz 1, 10587 Berlin, Germany
[2]Soil Science and Soil Protection, Martin Luther University Halle-Wittenberg, Von-Seckendorff-Platz 3, D-06120 Halle (Saale), Germany
[3]Laboratorio de Ecología Tropical Natural y Aplicada, Universidad Estatal Amazónica, Campus Principal Km 2.1/2 via a Napo (Paso Lateral) Puyo, Pastaza, Ecuador

*Correspondence to:* Antonia Zieger (antonia.zieger@tu-berlin.de)

Dear associate editor Yakov Kuzyakov, dear reviewers,

we would like to thank all reviewers for taking the time to provide us with your feedback on our manuscript. We have carefully considered your comments. Our responses and suggestions for possible changes are given below each comment.

5   Implementing these changes based on the comments of the reviewers has improved our manuscript. The reviewer comments have led us to clarify the abstract, the bulk density measurements, results of the bulk organic carbon and the data analysis. We included a paragraph about organic carbon stocks in section 4.3. Furthermore we improved the discussion on organic carbon response to the sawdust input. We feel that our suggested revisions will improve our manuscript beyond the level necessary to be published in Biogeo-sciences.

**Reviewer 2**

The authors have revised the manuscript very well. I have only some minor comments.

1) In the Abstract, please revise it in a concise way. For example, the main result is in the line 15, only a sentence.

15   *Author response: We summarised the main result in one sentence at the end of the abstract.*

2) Please give the full name of 'WRB' in page 4 Line22.
*Author response: We provided the full soil names according to the World Reference Base for Soil Resources (2015) in the Materials section.*

3) In the Table 1, please give full information of 'Xox' in the table title in Page 5. Second, what is the unit for Alox and other mineral, g kg-1 dry soil? or g kg-1 organic C?

*Author response: We gave full reference to $Al_{ox}$, $Si_{ox}$ and $Fe_{ox}$ in the table title instead of referring to $X_{ox}$. The oxalate-extractable metals are presented as $g\,kg^{-1}$ mineral part (or inorganic part) of the dry soil. We used this unit in order to evaluate the amount of oxalate-extractable metals in relation to total mineral constituents. The large and strongly varying concentrations of organic matter with depth masks the actual proportion of oxalate extractable minerals. For better comparability, we normalized the oxalate-extractable metals to the mineral soil component instead of to dry soil. For clarification, we added explanations to section "2.5. Acid oxalate extraction..." and the table titles.*

4) The similar description as above mentioned for Table 2 in Page 9.

*Author response: see above (3).*

5) In the result section, '3.2 sequential density fractionation' in Page 13 and Page 14, the description is not clear, please revise it in a concise way.

*Author response: We assume that you find section 3.2.1 (p. 13 line 3 to p. 14 line 2) not clear. We added additional information on the purpose of the data for each of the three paragraphs.*

6) In the figure 2, I am wondering why the sampling depths at the two sites are different?

*Author response: We sampled in the middle of each horizon. Horizon thickness varied slightly between profiles, which is a common feature of forest soils. In result also the sampling depths for the profiles selected for density fractionation were not identical.*

7) In Page 18, the methods and calculation description need to put in the material section? I am wondering why.

*Author response: We put the calculation description in the discussion, because we developed this new calculation method for evaluating the prevailing short range-ordered phases. The formulas used are based on well known formulas and assumptions. The assumptions result from the data of the oxalate-extraction. The idea to calculate the proportion of imogolite-type on short range-odered on the basis of solely aluminium is new. We revised section "4.2 Mineral composion..." to point this out.*

**Reviewer 3**

The manuscript brings interesting conclusions about the carbon storage capacity of andosols. The analyzes are well documented and lead to the description of interesting mechanisms to explain an important carbon storage capacity in the subsoil when the topsoil binding capacities are exhausted.

The manuscript would need a thorough English proofreading.

*Author response: A professional English proofreading was carried out.*

1) The authors could reformulate the title to state the positive result first (C storage in subsoils).

*Author response: We changed the title to "Massive carbon addition to an organic-rich Andosol increased the subsoil but not the topsoil carbon stock."*

2) Page 2, L.3: "Soil holds more organic carbon (OC) than there is carbon in the global vegetation and atmosphere combined": the authors should provide reference(s) from scientific literature.

*Author response: We provided the reference: Lehmann, Johannes, and Markus Kleber. "The Contentious Nature of Soil Organic Matter." Nature 528, no. 7580 (Dezember 2015): 60–68. https://doi.org/10.1038/nature16069.*

3) Page 2, L. 12: "the main factor": the authors should be more specific (of what?).

*Author response: We changed the sentence to: "Paustian et al. (1997) consider the OC stocks to increase linearly and limitless with increasing organic input."*

4) Page 4, L. 9-10: the authors could discuss the fact that higher vegetation in the sawdust site may have consequences on carbon intakes from the soil. Could the fact that vegetation is higher be due to sawdust additions?

*Author response: Unfortunately, we have no data of possible differences in plant biomass and plant community composition. We calculated a worst case scenario to exclude possible differences in C input with litter between the sites. The scenario shows, that differences in input of plant-derived C between sites are insignificant relative to the sawdust input. We clarified the purpose of the scenario calculation in the manuscript.*

5) Page 4, L. 16-19: a schema or photo of a soil profile would help to understand how horizons were delimited. It could be added as an appendix.

*Author response: We provided a picture of the soil profile in the Appendix.*

6) Page 5, Table 1: For each element presented in the table, the number of data that were pooled to calculate the mean value could be specified in the table (eg :"BD, g.cm-3, n=2" or "C/N, n=5). Standard deviations (rather than standard errors) should probably be used in tables. In column BD, why isn't the standard error/deviation specified?

*Author response: We added a row in the Table 1 showing the number of data that were pooled. We added the standard error of BD. According to Altman and Bland (2005) and Reinhart (2015), the standard deviation (SD) is a measure of variability and describes the spread of individual data points. The standard error (SE = SD/sqrt(n)) is a type of standard derivation, which describes how far the average for this sample might be from the true average. As we are more interested in the latter, we choose the standard error.*

*Altman, Douglas G, and J Martin Bland. "Standard Deviations and Standard Errors." British Medical Journal 331, no. 7521 (October 15, 2005): 903.*

*Reinhart, Alex. Statistics Done Wrong: The Woefully Complete Guide. No Starch Press, 2015.*

7) Page 9, Table 2 / Page 11, L.7: How was the "representative" profile chosen?

*Author response: The representative soil profiles were selected to meet the following criteria: five horizons within the upper 1 m, largest OC concentration in the topmost horizon, similar amounts of acid oxalate-extractable Al, Fe, and Si, and having different bulk OC concentrations in the third horizon. This was already lined out in section "2.7 Data analysis". We moved the*

10   *decision basis to the caption of Table 2.*

8) Page 13, L. 3 / Page 14, L.5: The authors should explain how values can exceed 100 wt%.

*Author response: We added the following sentence to section "3.2.1 Organic matter".: "Values exceeding 100 wt% are probably caused by random errors of measurements and some sodium polytungstate not entirely removed by sample washing."*

9) Page 13, L. 5-6 / Page 14, L.6: Could the authors explain why normalizations were performed?

*Author response: The amount of OC of the density fractionation are normalized to the mass of the fraction or the mass of the bulk soil representing either the OC concentration "in the fraction" or "with the fraction".*

20   10) Paragraph 4.4: The authors should emphasize more on the fact that the very low number of data used for comparing the sites may also be responsible for the absence of significative differences.

*Author response: We agree with you and included the calculation of statistical power and its results to the manuscripts discussion and Appendix.*

25   11) Page 23,L.30: The authors could add a sentence on the potential of mineral phases for carbon storage.

*Author response: We calculated the carbon stock potential for the case, that the Andosols upper horizon will one day stretch down to 100 cm and added a sentence in the conclusion accordingly.*

12) Page 24, L. 5-7: the absence of significative difference could also be due to the fact that the number of data was to small
30   to detect it.

*Author response: We emphasized this accordingly in the Discussion section 4.4 and the Conclusion.*

**Reviewer 4**

The manuscript answers the specific question of how much organic carbon (OC) can be stored in a specific type of soil (Andosol), giving experimental support to the theory of a finite OC accumulation potential of soils due to the limited binding capacities of minerals. I found the manuscript well written and well organized. Even though the experimental site was not de-
5  signed from the beginning for this trial, I agree with the authors saying that it is rare to have such a long history of treatments, and this is particularly valuable when studying carbon dynamics in soil. My bigger concern regards the bulk density (BD) values reported in table 1 and the way they were assessed. As reported in equation 1, BD is of a pivotal importance in assessing the total amount of OC in the soil (Mg/ha), so I think it should be specified with more details how the sampling was carried out, which volume of soil was considered, which corer and so on. Ten sampling points don't represent a huge number, but for
10  BD just 2 were used, which is really low for such a big area.

*Author response: We agree that the BD is of pivotal importance. We added the standard error in Table 1. The bulk density was determined at two profiles per site (all horizons). For each horizon, five replicates were sampled with $100\,cm^3$ corers, oven-dried at $105\,\circ C$ for 24 hours and weighted. We think that the number of bulk density measurements are sufficient, because*
15  *the standard error was small (with $<0.05\,g\,cm^{-3}$). We added those details in the revised manuscript.*

This observation arises because the BD values reported in table one are really low even for andosols ($<0.4\,Mg\,m^{-3}$), which causes a relatively low amount of OC per hectare considering the relevant percentage that OC reaches in that soil (up to 21%). For instance, $315\,Mg\,ha^{-1}$ found in this study (considering 1 m soil depth) are obtainable with a 2.6% of OC in a soil with BD
20  $= 1.2\,Mg\,m^{-3}$, very typical in mineral soils. Thus, besides improving the description of the sampling methodology, I suggest adding some reference supporting such low BD values.

*Author response: We added some references supporting BD values below $0.5\,g\,cm^{-3}$ in Andosols. Especially aluandic Andosols feature often such low BD values. We also added some sentences to Section 4.3 to compare the OC stock of this Andosol*
25  *with values from other soils.*

*Basile-Doelsch, I., R. Amundson, W.E.E. Stone, D. Borschneck, J. Y. Bottero, S. Moustier, F. Masin, and F. Colin. "Mineral Control of Carbon Pools in a Volcanic Soil Horizon." Geoderma 137, no. 3-4 (2007): 477-489. https://doi.org/10.1016/j.geoderma.2006.10.006.*

*Marin-Spiotta, Erika, Oliver A. Chadwick, Marc Kramer, and Mariah S. Carbone. "Carbon Delivery to Deep Mineral Horizons in Hawaiian Rain Forest Soils." Journal of Geophysical Research: Biogeosciences 116, no. 63 (2011): https://doi.org/10.1029/2010JG001587.*

*Tonneijck, Femke H., B. Jansen, K. G. J. Nierop, Jacobus M. Verstraten, Jan Sevink, and L. de Lange. "Towards Understanding of Carbon Stocks and Stabilization in Volcanic Ash Soils in Natural Andean Ecosystems of Northern Ecuador." European Journal of Soil Science 61, no. 3 (2010): 392-405. https://doi.org/10.1111/j.13652389.2010.01241.x.*

5  I would also stress this fact more in the discussion chapter, section 4.4: the low BD is likely due to a low content of minerals, which can be "easily" saturated with OC at least in the upper layers. Please provide also some more info about the way in which sawdust was distributed over the years, specifying to which degree can the soil treated with sawdust considered homogeneous within the 3 hectares of the trial.

10  *Author response: Yes, the smaller content of mineral phases in the topsoil might cause the smaller sorption capacity but probably the smaller mineral content itself is just a result of the strong sorption of organic matter by highly reactive mineral phases. The sawdust was distributed approximately evenly over the site by hand. We added this information to the Materials and Methods section 2.1.*

15  Some technical notes:

P1-L3: please add a reference at the end of the first sentence (storage of organic carbon in soils)

P4-L11: please use the same decimals (1-16 or 0.9-16.0 $\mathrm{Mg\,ha^{-1}\ year^{-1}}$)

P6-L9: standard deviation (not derivation)

*Author response: Thank you for pointing this out. We changed the manuscript accordingly.*

With kind regards

Antonia Zieger on behalf of all of the co-authors

[revised manuscript text omitted]